# Predictors of Urgent Cancer Care Clinic and Emergency Department Visits for Individuals Diagnosed with Cancer

Kathleen Decker [1,2,3,*], Pascal Lambert [1,3], Katie Galloway [3], Oliver Bucher [3], Marshall Pitz [1,4,5], Benjamin Goldenberg [4,5], Harminder Singh [1,5], Mark Kristjanson [6], Tunji Fatoye [6,7] and Eric J. Bow [4,5,8]

1   CancerCare Manitoba Research Institute, 675 McDermot Avenue, CancerCare Manitoba, Winnipeg, MB R3E 0V9, Canada; plambert@cancercare.mb.ca (P.L.); mpitz@cancercare.mb.ca (M.P.); harminder.singh@umanitoba.ca (H.S.)
2   Department of Community Health Sciences, University of Manitoba, 750 Bannatyne Avenue, Winnipeg, MB R3E 0V9, Canada
3   Department of Epidemiology and Cancer Registry, CancerCare Manitoba, 675 McDermot Avenue, Winnipeg, MB R3E 0V9, Canada; kgalloway@cancercare.mb.ca (K.G.); obucher@cancercare.mb.ca (O.B.)
4   Department of Medical Oncology and Hematology, CancerCare Manitoba, 675 McDermot Avenue, Winnipeg, MB R3E 0V9, Canada; bgoldenberg@cancercare.mb.ca (B.G.); ebow@cancercare.mb.ca (E.J.B.)
5   Department of Internal Medicine, University of Manitoba, 820 Sherbrook Street, Winnipeg, MB R3E 0V9, Canada
6   Department of Primary Care Oncology, Cancer Care Manitoba, 675 McDermot Avenue, Winnipeg, MB R3E 0V9, Canada; mkristjanson@cancercare.mb.ca (M.K.); tfatoye@sogh.mb.ca (T.F.)
7   Department of Family Medicine, University of Manitoba, 750 Bannatyne Avenue, Winnipeg, MB R3E 0V9, Canada
8   Department of Medical Microbiology and Infectious Diseases, University of Manitoba, 745 Bannatyne Avenue, Winnipeg, MB R3E 0V9, Canada
*   Correspondence: kdecker@cancercare.mb.ca; Tel.: +1-204-787-4933

**Abstract:** In 2013, CancerCare Manitoba (CCMB) launched an urgent cancer care clinic (UCC) to meet the needs of individuals diagnosed with cancer experiencing acute complications of cancer or its treatment. This retrospective cohort study compared the characteristics of individuals diagnosed with cancer that visited the UCC to those who visited an emergency department (ED) and determined predictors of use. Multivariable logistic mixed models were run to predict an individual's likelihood of visiting the UCC or an ED. Scaled Brier scores were calculated to determine how greatly each predictor impacted UCC or ED use. We found that UCC visits increased up to 4 months after eligibility to visit and then decreased. ED visits were highest immediately after eligibility and then decreased. The median number of hours between triage and discharge was 2 h for UCC visits and 9 h for ED visits. Chemotherapy had the strongest association with UCC visits, whereas ED visits prior to diagnosis had the strongest association with ED visits. Variables related to socioeconomic status were less strongly associated with UCC or ED visits. Future studies would be beneficial to planning service delivery and improving clinical outcomes and patient satisfaction.

**Keywords:** oncology; urgent care clinics; emergency service



## 1. Introduction

With the increasing number of cancer cases in the population and the widespread use of newer, more effective out-patient systemic and radiation therapy (RT), emergency departments (EDs) are seeing more individuals diagnosed with cancer who present with symptoms related to the effects of treatment as well as their underlying disease [1–5]. The most common reasons for ED use by oncology patients are for fatigue, pain, fever, dyspnea, and infection-related syndromes [6–9]. However, EDs may not always be the most appropriate care setting for this population. EDs are often overcrowded, provide care to a wide spectrum of patients, and have long wait times [2,10]. Many ED visits by

individuals during chemotherapy occur during daytime hours and could be addressed in a non-ED setting [7,11,12].

An alternative option to EDs for individuals that need acute care for cancer or treatment-related issues is an urgent cancer care clinic (UCC) located within the cancer treatment facility. The advantages of providing care through a UCC may include increased convenience for individuals diagnosed with cancer and their family members, which in turn may decrease delays in seeking care, more timely care, and specialized care because UCCs can be staffed by health care providers who have specialized training in the care of individuals with cancer and the side-effects of cancer treatment. In November 2013, CancerCare Manitoba (CCMB) launched a UCC. The UCC aims to meet the needs of individuals diagnosed with cancer experiencing acute complications of cancer or its treatment, thereby providing an avenue for prompt contextual care, thereby reducing the need for patients to access hospital ED services. The objectives of this cohort study were to describe the characteristics of individuals diagnosed with cancer that used the UCC compared to those who used an ED and determine predictors of UCC and ED visits.

## 2. Materials and Methods

### 2.1. Setting

CCMB is the provincial cancer agency responsible for providing clinical services to all Manitobans diagnosed with cancer. The UCC is located at CCMB's main site in the city of Winnipeg's core area. During the study time period, there were 6 EDs located throughout Winnipeg, including an ED located at the Health Sciences Centre, Manitoba's largest academic healthcare facility.

The UCC operates from 8 a.m. to 4 p.m., Monday to Friday. Individuals can be referred to the UCC by their oncology team, primary care clinician (PCC), or by self-referral. Approximately 10 patients are seen at the UCC each day. To be eligible to visit the UCC, individuals must be receiving treatment (chemotherapy, radiation therapy (RT), immunotherapy, or hormone therapy (HT)) or follow-up care from a CCMB clinic/provider. Once an individual arrives at the UCC, they are registered, assessed and triaged, and then seen by a family physician with additional training in oncology (FPO). Further investigations may be conducted, and immediate treatment may be provided. Patients are discharged to their homes with follow-up by their CCMB clinic, to an outpatient hospital service, or to an in-patient hospital-based service.

### 2.2. Study Design and Population

The retrospective cohort study included individuals diagnosed with invasive cancer (excluding non-melanoma skin cancer) between 2008 and 2015 who were 18 years of age or older at diagnosis, lived in Winnipeg, and received treatment or follow-up care between 4 November 2013 and 31 December 2016 (eligible to visit the UCC). Entry into the study cohort depended on diagnosis date and when an individual became eligible to visit the UCC (Figure S1). Study follow-up time analyzed was relative to an individual's diagnosis date and follow-up time ended 1 year after that individual's last scheduled CCMB visit. Individuals required 2 or more years of continuous health coverage prior to diagnosis to be included in the study. Non-cancer related ED visits were excluded (Table S1). This study was approved by the University of Manitoba's Health Research Ethics Board, Manitoba Health's Health Information and Privacy Committee, and CCMB's Research and Resource Impact Committee.

### 2.3. Data Sources

The Manitoba Cancer Registry (MCR) was used to identify individuals diagnosed with cancer, their sex, birth date, diagnosis date, cancer site, stage, treatment, and postal code at diagnosis. The MCR is a population-based registry that is legally mandated to collect and maintain accurate, comprehensive information about cancer cases in Manitoba and has consistently been shown to be of very high quality [13]. The CCMB electronic

medical record (ARIA™) was used to identify UCC visits, Canadian Triage and Acuity Scale (CTAS) scores, and dates and times of triage, treatment, and discharge.

The Manitoba Population Registry contains demographic, vital status, and migration information and was used to determine provincial health coverage duration and residential mobility. The Medical Claims Database is generated by claims filed by healthcare providers for reimbursement of services and was used to determine primary care clinician (PCC) visits, continuity of care, and comorbidity. The Hospital Discharge Abstracts Database includes all hospital admissions for Manitoba residents and was used to determine comorbidity. The Drug Program Information Network database captures medication information from community pharmacies and was used to determine treatment provided by oral administration (i.e., oral systemic therapy). The accuracy and completeness of these administrative health databases have been established [14–16]. ED visits were identified using the Winnipeg Regional Health Authority's Admissions, Discharge and Transfer and E-Triage data and Emergency Department Information System databases [17]. Statistics Canada census data were used to determine area-level average household income based on each individual's postal code [18–20].

### 2.4. Definition of Variables

Each individual's cancer site was categorized as hematologic, genitourinary, lung and bronchus, breast, digestive, or other. Stage at diagnosis was classified using the American Joint Committee on Cancer collaborative staging system 7th edition (I, II, III, IV, unknown) [21]. Area-level average household income was categorized by quintile from Q1 (lowest) to Q5 (highest). Residential mobility was defined as high (3 or more postal code changes in the 5 years prior to diagnosis) or low (less than 3 postal code changes in the 5 years prior to diagnosis). Comorbidity was measured using the Johns Hopkins Adjusted Clinical Group System software Resource Utilization Band (RUB) (0: non-user, 1: healthy user, 2: low morbidity, 3: moderate morbidity, 4: high morbidity, 5: very high morbidity) [22]. Continuity of care was measured by determining the individuals with at least 50% of visits to the same PCC among those with at least 3 visits in 6 to 30 months prior to diagnosis (yes; ≥50%, no; <50%, <3 visits) [23]. Frequency of ED visits in the 3 months prior to diagnosis was categorized as 0, 1, or 2 or more. The 5-level CTAS score (level 1: resuscitation, level 2: emergent, level 3: urgent, level 4: less urgent, level 5: non-urgent) was used to assess the urgency of a patient's condition upon visiting the UCC or an ED (16). Treatment at the time of a UCC or ED visit was described as active (cancer treatment within 30 days of the visit), inactive (cancer treatment previously but not within 30 days of the visit), or no treatment.

### 2.5. Outcomes

The primary outcome was defined as a UCC or ED visit for a cancer-related reason.

### 2.6. Statistical Analysis

Descriptive statistics were used to characterize individuals who visited the UCC or ED. To predict an individual's likelihood of visiting the UCC or ED, multivariable logistic mixed models were run. Predictors included treatment variables, diagnostic characteristics, sociodemographic characteristics, and healthcare use history. Due to long follow-up times, interaction terms between predictors and the start of follow-up time were included as the average effect over time could potentially hide the varying effect of predictors over time. However, limited power led to model non-convergence. Therefore, analyses were stratified by follow-up time after the start of eligibility to attend the UCC (1–6 months, 7–12 months, 13–18 months, and 19–24 months). Natural splines were used for continuous predictors if they demonstrated a non-linear relationship with an outcome. Non-linear relationships were described with plots of predicted probabilities with covariates held at their median for continuous variables and proportion for categorical variables. To account for time-varying predictors and incomplete follow-up during an interval, an offset of log-

time during monthly intervals was included. Sensitivity analyses were performed without sex in the model, including ED visits that occurred only during UCC's hours of operation and ED visits regardless of visit reason. Logistic mixed models were run using the R package GLMMadaptive [24]. Estimates were marginalized to obtain population-averaged output [25].

To determine how greatly each predictor impacted UCC or ED use, scaled Brier scores were calculated by squaring the difference between predicted and outcome values and scaled to the outcome prevalence in the cohort, where 1 indicated perfect prediction, 0 indicated a random association, and a negative value indicated predictions worse than chance [26,27]. The percentage increase between the scaled Brier scores from the full model minus a predictor and full model were calculated and plotted. Higher percentage increases indicated a higher impact of the predictor on the outcome.

## 3. Results

### 3.1. Characteristics of Individuals Who Visited the UCC or an ED

From 4 November 2013 to 31 December 2016, there were 3152 visits to the UCC and 10,100 visits to an ED by individuals who were receiving treatment or follow-up care from CCMB. A total of 4195 ED visits (41.5%) occurred during UCC hours of operation. Table 1 describes the characteristics of individuals who visited the UCC or an ED by treatment-related variables, cancer-related variables, socio-demographic variables, and health care utilization variables. A higher percentage of individuals who visited the UCC were female, had a higher income, lower residential mobility, lower comorbidity, a stage IV diagnosis, and an unknown CTAS score and were receiving treatment at the time of their visit compared to individuals who visited an ED. The characteristics of individuals who had an ED visit during UCC hours of operation and those who had an ED visit outside these times were similar. Of the individuals who had an ED visit, 4 to 14% also had a UCC visit. Of the individuals who had a UCC visit, 18 to 52% also had an ED visit.

**Table 1.** Comparison of individuals who visited the Urgent Cancer Care Clinic (UCC) or an emergency department (ED) by follow-up period, 2013–2016, Winnipeg, Manitoba (*N* = 13,252 visits).

| Variables | UCC Visits, No. (%) | | | | ED Visits, No. (%) | | | |
|---|---|---|---|---|---|---|---|---|
| | Follow-Up Period | | | | Follow-Up Period | | | |
| | 1 to 6 (*N* = 1358) | 7 to 12 (*N* = 959) | 13 to 18 (*N* = 531) | 19 to 24 (*N* = 304) | 1 to 6 (*N* = 4203) | 7 to 12 (*N* = 2491) | 13 to 18 (*N* = 1943) | 19 to 24 (*N* = 1463) |
| Chemotherapy status at time of visit | | | | | | | | |
| Active | 921 (68) | 485 (51) | 178 (34) | 109 (36) | 1147 (27) | 669 (27) | 282 (15) | 176 (12) |
| Inactive | 93 (7) | 330 (34) | 250 (47) | 136 (45) | 180 (4) | 590 (24) | 711 (37) | 575 (39) |
| None | 344 (25) | 144 (15) | 103 (19) | 59 (19) | 2876 (68) | 1232 (49) | 950 (49) | 712 (49) |
| Radiation therapy status at time of visit | | | | | | | | |
| Active | 305 (22) | 163 (17) | 43 (8) | 13 (4) | 423 (10) | 137 (6) | 63 (3) | 43 (3) |
| Inactive | 105 (8) | 273 (28) | 212 (40) | 142 (47) | 280 (7) | 730 (29) | 674 (35) | 554 (38) |
| None | 948 (70) | 523 (55) | 276 (52) | 149 (49) | 3500 (83) | 1624 (65) | 1206 (62) | 866 (59) |
| Hormone therapy status at time of visit | | | | | | | | |
| Active | 35 (2) | 80 (8) | 37 (7) | 16 (5) | 115 (3) | 142 (6) | 142 (7) | 116 (8) |
| Inactive | 12 (1) | 42 (4) | 25 (5) | 23 (8) | 70 (2) | 184 (7) | 181 (9) | 199 (14) |
| None | 1314 (97) | 837 (87) | 469 (88) | 265 (87) | 4018 (96) | 2165 (87) | 1620 (83) | 1148 (78) |
| Immunotherapy status at time of visit | | | | | | | | |
| Active | 175 (13) | 115 (12) | 66 (12) | 29 (10) | 191 (5) | 111 (4) | 97 (5) | 52 (4) |
| Inactive | 18 (1) | 50 (5) | 70 (13) | 60 (20) | 21 (1) | 89 (4) | 118 (6) | 121 (8) |
| None | 1165 (86) | 794 (83) | 395 (74) | 215 (71) | 3991 (95) | 2291 (92) | 1728 (89) | 1290 (88) |

**Table 1.** *Cont.*

| Variables | UCC Visits, No. (%) | | | | ED Visits, No. (%) | | | |
|---|---|---|---|---|---|---|---|---|
| | Follow-Up Period | | | | Follow-Up Period | | | |
| | 1 to 6 (*N* = 1358) | 7 to 12 (*N* = 959) | 13 to 18 (*N* = 531) | 19 to 24 (*N* = 304) | 1 to 6 (*N* = 4203) | 7 to 12 (*N* = 2491) | 13 to 18 (*N* = 1943) | 19 to 24 (*N* = 1463) |
| Site | | | | | | | | |
| Hematologic | 299 (22) | 177 (18) | 123 (23) | 76 (25) | 535 (13) | 295 (12) | 245 (13) | 176 (12) |
| Genitourinary | 148 (11) | 110 (11) | 96 (18) | 78 (26) | 677 (16) | 518 (21) | 411 (21) | 364 (25) |
| Lung and bronchus | 215 (16) | 173 (18) | 84 (16) | 34 (11) | 807 (19) | 333 (13) | 239 (12) | 156 (11) |
| Breast | 280 (20) | 239 (25) | 56 (11) | 34 (11) | 487 (11) | 339 (14) | 260 (13) | 218 (15) |
| Digestive | 279 (21) | 171 (18) | 86 (16) | 49 (16) | 1170 (28) | 666 (27) | 493 (25) | 337 (23) |
| Other | 137 (10) | 89 (9) | 86 (16) | 33 (11) | 527 (13) | 340 (14) | 295 (15) | 212 (14) |
| Stage | | | | | | | | |
| I | 124 (9) | 107 (11) | 46 (9) | 25 (8) | 585 (14) | 426 (17) | 376 (19) | 324 (22) |
| II | 256 (19) | 229 (24) | 85 (16) | 82 (27) | 728 (17) | 557 (22) | 471 (24) | 378 (26) |
| III | 345 (25) | 240 (25) | 122 (23) | 70 (23) | 818 (19) | 562 (23) | 397 (20) | 326 (22) |
| IV | 433 (32) | 259 (27) | 168 (32) | 63 (21) | 1437 (34) | 620 (25) | 394 (20) | 242 (17) |
| Unknown | 200 (15) | 124 (13) | 110 (21) | 64 (21) | 635 (15) | 326 (13) | 305 (16) | 193 (13) |
| Age at diagnosis (mean (SD)) | 61 (12) | 62 (12) | 62 (12) | 60 (13) | 67 (14) | 66 (14) | 66 (14) | 67 (14) |
| Sex | | | | | | | | |
| Male | 550 (41) | 345 (36) | 286 (54) | 140 (46) | 2036 (48) | 1272 (51) | 1003 (52) | 677 (46) |
| Female | 808 (60) | 614 (64) | 245 (46) | 164 (54) | 2167 (52) | 1219 (49) | 940 (48) | 786 (54) |
| Income quintile (Q) | | | | | | | | |
| Q1 (lowest) | 282 (21) | 180 (19) | 87 (16) | 53 (17) | 1147 (25) | 663 (27) | 482 (25) | 339 (23) |
| Q2 | 264 (19) | 198 (21) | 88 (17) | 62 (20) | 803 (19) | 509 (20) | 383 (20) | 332 (23) |
| Q3 | 294 (22) | 185 (19) | 128 (24) | 82 (27) | 850 (20) | 460 (18) | 364 (19) | 292 (20) |
| Q4 | 286 (21) | 218 (23) | 128 (24) | 44 (14) | 777 (18) | 457 (18) | 407 (21) | 282 (19) |
| Q5 (highest) | 232 (17) | 178 (19) | 100 (19) | 63 (21) | 726 (17) | 402 (16) | 307 (16) | 218 (15) |
| Residential mobility [a] | | | | | | | | |
| Low | 1268 (93) | 903 (94) | 497 (94) | 264 (87) | 3819 (91) | 2283 (92) | 1780 (92) | 1338 (91) |
| High | 90 (7) | 56 (6) | 34 (6) | 40 (13) | 384 (9) | 208 (8) | 163 (8) | 125 (9) |
| Continuity of care [b] | | | | | | | | |
| Yes (≥50%) | 936 (69) | 648 (68) | 364 (69) | 221 (73) | 2993 (71) | 1761 (71) | 1336 (69) | 1024 (70) |
| No (<50%) | 312 (23) | 230 (24) | 119 (22) | 54 (18) | 854 (20) | 553 (22) | 445 (23) | 314 (21) |
| <3 visits | 110 (8) | 81 (8) | 48 (9) | 29 (10) | 356 (8) | 177 (7) | 162 (8) | 125 (9) |
| Emergency department visits in 3 months prior to diagnosis | | | | | | | | |
| 0 | 847 (62) | 634 (66) | 361 (68) | 204 (67) | 2243 (53) | 1507 (61) | 1172 (60) | 963 (66) |
| 1 | 288 (21) | 212 (22) | 106 (20) | 68 (22) | 1100 (26) | 571 (23) | 472 (24) | 281 (19) |
| 2 or more | 223 (16) | 113 (12) | 64 (12) | 32 (11) | 860 (20) | 413 (17) | 299 (15) | 219 (15) |

**Table 1.** *Cont.*

| Variables | UCC Visits, No. (%) | | | | ED Visits, No. (%) | | | |
|---|---|---|---|---|---|---|---|---|
| | Follow-Up Period | | | | Follow-Up Period | | | |
| | 1 to 6 (*N* = 1358) | 7 to 12 (*N* = 959) | 13 to 18 (*N* = 531) | 19 to 24 (*N* = 304) | 1 to 6 (*N* = 4203) | 7 to 12 (*N* = 2491) | 13 to 18 (*N* = 1943) | 19 to 24 (*N* = 1463) |
| Primary care clinician visits | | | | | | | | |
| Median (IQR) | 6 (3–10) | 6 (3–10) | 6 (3–10) | 6 (3–10) | 7 (3–12) | 7 (4–12) | 7 (4–13) | 8 (4–14) |
| Comorbidity (resource use band) | | | | | | | | |
| 0, 1, 2 | 288 (21) | 208 (22) | 101 (19) | 55 (18) | 740 (18) | 395 (16) | 321 (17) | 235 (16) |
| 3 | 810 (60) | 585 (61) | 318 (60) | 186 (61) | 2377 (57) | 1406 (56) | 1096 (56) | 788 (54) |
| 4, 5 | 260 (19) | 166 (17) | 112 (21) | 63 (21) | 1086 (26) | 690 (28) | 526 (27) | 440 (30) |
| Canadian Triage Acuity Score (CTAS) | | | | | | | | |
| 1, 2 | 338 (25) | 197 (21) | 94 (18) | 54 (18) | 1081 (26) | 674 (27) | 495 (25) | 361 (25) |
| 3, 4, 5 | 914 (67) | 706 (74) | 403 (76) | 229 (75) | 3115 (74) | 1814 (73) | 1441 (74) | 1098 (75) |
| Unknown | 106 (8) | 56 (6) | 34 (6) | 21 (7) | 7 (0) | 3 (0) | 7 (0) | 4 (0) |
| Days between diagnosis and ED/UCC visit | | | | | | | | |
| Median (IQR) | 105 (67–140) | 249 (211–295) | 428 (394–483) | 615 (576–670) | 70 (27–121) | 263 (219–307) | 443 (400–493) | 624 (580–674) |
| Days between diagnosis and start of eligibility | | | | | | | | |
| Median (IQR) | 0 (0–0) | 0 (0–0) | 0 (0–0) | 0 (0–0) | 0 (0–0) | 0 (0–0) | 0 (0–0) | 0 (0–0) |
| Hours between triage and discharge | | | | | | | | |
| Median (IQR) | 2 (1–3) | 2 (1–3) | 2 (1–3) | 2 (1–3) | 9 (5–20) | 9 (5–19) | 9 (5–19) | 8 (5–18) |
| Hours between triage and exam room | | | | | | | | |
| Median (IQR) | 1 (1–2) | 1 (1–2) | 1 (1–2) | 1 (1–2) | NA | NA | NA | NA |

Abbreviations: UCC, urgent cancer care; ED, emergency department; IQR, interquartile range; NA, not available. [a] Residential mobility included postal code changes in the 5 years prior to diagnosis. [b] Continuity of care included primary care clinician visits in the 6 to 30 months prior to diagnosis.

### 3.2. Visit Length of Time

The median number of hours between triage and discharge for UCC visits was 2 h (inter-quartile range (IQR) 1–3). The median number of hours between triage and discharge for ED visits was 9 h (IQR 5–19). The number of hours between triage and arriving in the exam room was available for 2533 UCC visits (80.4%) but for no ED visits. The median time was 1 h (IQR 1–2) with little variability by follow-up time period. The mean and median number of hours were higher for individuals with a CTAS score of 1 (resuscitation) or 2 (emergent) (Table S2).

### 3.3. Rate of Visits

The rate of UCC visits increased during the first 4 months after the start of UCC eligibility and then decreased over time (Figure 1). In contrast, the rate of ED visits was highest immediately after the start of eligibility and decreased over time. The ED visit rate was 4 times higher than the UCC visit rate.

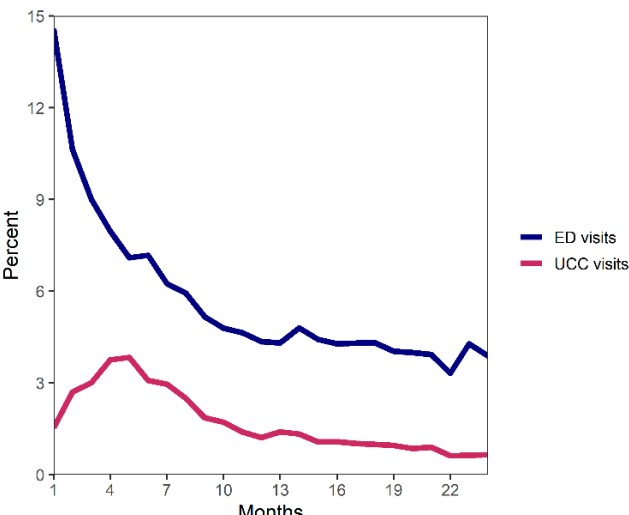

**Figure 1.** Percentage of individuals diagnosed with cancer that had a UCC or ED visit by number of months since the start of eligibility time frame (3 months prior to first scheduled CancerCare Manitoba (CCMB) visit or, if first CCMB visit occurred prior to diagnosis, date of diagnosis).

### 3.4. Association between Predictors and UCC Visits

Treatment variables: Chemotherapy was strongly associated with UCC visits across all follow-up time periods (Figure 2; Table S3). Individuals who were being treated with chemotherapy were significantly more likely to visit the UCC compared to those who were not receiving chemotherapy in all follow-up periods (Table 2).

**Table 2.** Multivariable logistic regression model describing factors associated with UCC visits stratified by follow-up time period.

| Variable | Follow-Up Period | | | | | | | | | | | |
|---|---|---|---|---|---|---|---|---|---|---|---|---|
| | **1 to 6 Months** | | | **7 to 12 Months** | | | **13 to 18 Months** | | | **19 to 24 Months** | | |
| | **OR** | **95% CI** | ***p*** | **OR** | **95% CI** | ***p*** | **OR** | **95% CI** | ***p*** | **OR** | **95% CI** | ***p*** |
| Chemotherapy | | | | | | | | | | | | |
| Inactive | 1.00 | | | 1.00 | | | 1.00 | | | 1.00 | | |
| Active | 3.08 | 2.32–4.09 | <0.01 | 2.80 | 2.29–3.43 | <0.01 | 2.62 | 1.98–3.47 | <0.01 | 4.65 | 3.38–6.41 | <0.01 |
| None | 0.45 | 0.33–0.62 | | 0.39 | 0.30–0.51 | | 0.41 | 0.29–0.59 | | 0.48 | 0.31–0.74 | |
| Radiation Therapy | | | | | | | | | | | | |
| Inactive | 1.00 | | | 1.00 | | | 1.00 | | | 1.00 | | |
| Active | 2.21 | 1.62–3.00 | <0.01 | 1.46 | 1.13–1.89 | <0.01 | 1.79 | 1.18–2.73 | <0.01 | 1.08 | 0.52–2.22 | <0.01 |
| None | 1.06 | 0.79–1.42 | | 0.76 | 0.62–0.93 | | 0.74 | 0.56–0.97 | | 0.52 | 0.36–0.74 | |
| Hormone Therapy | | | | | | | | | | | | |
| Inactive | 1.00 | | | 1.00 | | | 1.00 | | | 1.00 | | |
| Active | 1.58 | 0.75–3.32 | <0.01 | 1.39 | 0.87–2.20 | <0.01 | 1.49 | 0.77–2.87 | <0.01 | 1.00 | 0.46–2.16 | 0.14 |
| None | 3.01 | 1.62–5.58 | | 2.38 | 1.61–3.51 | | 3.48 | 1.94–6.24 | | 1.99 | 0.96–4.09 | |

**Table 2.** *Cont.*

| Variable | 1 to 6 Months | | | 7 to 12 Months | | | 13 to 18 Months | | | 19 to 24 Months | | |
|---|---|---|---|---|---|---|---|---|---|---|---|---|
| | **OR** | **95% CI** | *p* | **OR** | **95% CI** | *p* | **OR** | **95% CI** | *p* | **OR** | **95% CI** | *p* |
| Immunotherapy | | | | | | | | | | | | |
| Inactive | 1.00 | | | 1.00 | | | 1.00 | | | 1.00 | | |
| Active | 1.83 | 0.92–3.65 | 0.22 | 1.06 | 0.67–1.67 | 0.93 | 1.30 | 0.86–1.98 | <0.01 | 0.72 | 0.41–1.25 | <0.01 |
| None | 1.76 | 0.88–3.54 | | 1.00 | 0.61–1.65 | | 0.66 | 0.45–0.98 | | 0.44 | 0.26–0.74 | |
| Cancer Site | | | | | | | | | | | | |
| Hematologic | 1.78 | 1.31–2.42 | | 1.84 | 1.22–2.78 | | 1.20 | 0.80–1.80 | | 1.68 | 0.96–2.94 | |
| Genitourinary | 1.02 | 0.76–1.39 | | 0.94 | 0.60–1.50 | | 1.54 | 0.97–2.46 | | 2.54 | 1.30–4.99 | |
| Lung and bronchus | 1.51 | 1.06–2.15 | <0.01 | 1.93 | 1.30–2.86 | <0.01 | 1.93 | 1.12–3.32 | <0.01 | 3.43 | 1.56–7.58 | <0.01 |
| Breast | 2.09 | 1.44–3.03 | | 1.65 | 1.09–2.50 | | 1.02 | 0.51–2.04 | | 0.73 | 0.30–1.75 | |
| Digestive | 1.14 | 0.82–1.59 | | 0.82 | 0.54–1.26 | | 0.75 | 0.46–1.21 | | 1.42 | 0.77–2.62 | |
| Other | 1.00 | | | 1.00 | | | 1.00 | | | 1.00 | | |
| Stage | | | | | | | | | | | | |
| I | 1.00 | | | 1.00 | | | 1.00 | | | 1.00 | | |
| II | 1.44 | 1.08–1.93 | | 1.78 | 1.27–2.50 | | 1.73 | 1.10–2.71 | | 3.24 | 1.66–6.31 | |
| III | 1.75 | 1.29–2.39 | <0.01 | 1.78 | 1.27–2.51 | <0.01 | 2.17 | 1.33–3.55 | <0.01 | 2.62 | 1.35–5.07 | <0.01 |
| IV | 2.65 | 1.94–3.61 | | 2.38 | 1.59–3.54 | | 3.85 | 2.39–6.21 | | 3.12 | 1.53–6.36 | |
| Unknown | 1.84 | 1.18–2.85 | | 1.44 | 0.89–2.34 | | 3.71 | 2.14–6.41 | | 5.94 | 2.80–12.58 | |
| Diagnosis age [a] | | | | | | | | | | | | |
| ' | 0.59 | 0.41–0.87 | <0.01 | 1.03 | 0.61–1.73 | <0.01 | 0.80 | 0.43–1.48 | <0.01 | 0.86 | 0.75–0.98 | 0.02 |
| '' | 0.42 | 0.31–0.56 | | 0.56 | 0.37–0.84 | | 0.43 | 0.27–0.69 | | | | |
| Sex | | | | | | | | | | | | |
| Female | 1.00 | | | 1.00 | | | 1.00 | | | 1.00 | | |
| Male | 0.84 | 0.71–1.00 | 0.05 | 0.81 | 0.66–1.00 | 0.05 | 1.27 | 0.99–1.63 | 0.07 | 0.86 | 0.58–1.28 | 0.46 |
| Income quintile | | | | | | | | | | | | |
| Q1 | 1.42 | 1.12–1.80 | | 1.21 | 0.90–1.62 | | 0.98 | 0.69–1.39 | | 1.07 | 0.64–1.80 | |
| Q2 | 1.27 | 0.97–1.67 | | 1.23 | 0.91–1.66 | | 1.18 | 0.77–1.79 | | 0.90 | 0.53–1.52 | |
| Q3 | 1.35 | 1.04–1.74 | 0.05 | 1.07 | 0.82–1.38 | 0.64 | 1.63 | 1.14–2.33 | 0.04 | 1.41 | 0.85–2.36 | 0.18 |
| Q4 | 1.18 | 0.92–1.51 | | 1.15 | 0.87–1.53 | | 1.21 | 0.78–1.87 | | 0.85 | 0.50–1.43 | |
| Q5 | 1.00 | | | 1.00 | | | 1.00 | | | 1.00 | | |
| Residential mobility [b] | | | | | | | | | | | | |
| Low | 1.00 | | | 1.00 | | | 1.00 | | | 1.00 | | |
| High | 0.91 | 0.66–1.25 | 0.56 | 1.02 | 0.71–1.46 | 0.92 | 1.16 | 0.71–1.90 | 0.56 | 1.57 | 0.90–2.72 | 0.11 |
| ED visits prior to diagnosis | | | | | | | | | | | | |
| 0 | 1.00 | | | 1.00 | | | 1.00 | | | 1.00 | | |
| 1 | 1.14 | 0.94–1.38 | <0.01 | 1.65 | 1.29–2.11 | <0.01 | 1.06 | 0.77–1.47 | 0.32 | 1.34 | 0.89–2.01 | 0.27 |
| 2+ | 1.51 | 1.20–1.91 | | 1.78 | 1.29–2.46 | | 1.37 | 0.91–2.05 | | 1.31 | 0.76–2.28 | |

**Table 2.** *Cont.*

| Variable | Follow-Up Period | | | | | | | | | | | |
|---|---|---|---|---|---|---|---|---|---|---|---|---|
| | 1 to 6 Months | | | 7 to 12 Months | | | 13 to 18 Months | | | 19 to 24 Months | | |
| | OR | 95% CI | *p* | OR | 95% CI | *p* | OR | 95% CI | *p* | OR | 95% CI | *p* |
| Primary care clinician visits | | | | | | | | | | | | |
| Per 10 visits | 0.99 | 0.89–1.11 | 0.92 | 1.06 | 0.94–1.18 | 0.34 | 0.96 | 0.79–1.17 | 0.71 | 1.05 | 0.86–1.28 | 0.64 |
| Continuity of care [c] | | | | | | | | | | | | |
| Yes (≥50%) | 1.00 | | | 1.00 | | | 1.00 | | | 1.00 | | |
| No (<50%) | 1.18 | 0.95–1.47 | 0.04 | 1.25 | 1.02–1.53 | 0.04 | 1.13 | 0.85–1.50 | 0.50 | 1.05 | 0.73–1.51 | 0.92 |
| <3 visits | 0.76 | 0.55–1.05 | | 0.80 | 0.56–1.13 | | 0.83 | 0.52–1.32 | | 0.91 | 0.45–1.86 | |
| Comorbidity (RUB) | | | | | | | | | | | | |
| 0–2 | 1.00 | | | 1.00 | | | 1.00 | | | 1.00 | | |
| 3 | 1.33 | 1.08–1.64 | <0.01 | 1.12 | 0.85–1.47 | 0.54 | 1.33 | 0.92–1.91 | 0.05 | 1.58 | 0.91–2.75 | 0.16 |
| 4–5 | 1.51 | 1.16–1.98 | | 1.22 | 0.85–1.76 | | 1.95 | 1.15–3.33 | | 1.86 | 0.98–3.52 | |
| Time Interval | | | | | | | | | | | | |
| ′ | 0.98 | 0.62–1.54 | 0.96 | 0.91 | 0.86–0.96 | <0.01 | 0.94 | 0.88–1.00 | 0.05 | 0.93 | 0.86–0.99 | 0.03 |
| ″ | 0.97 | 0.80–1.18 | | | | | | | | | | |

Abbreviations: UCC, urgent cancer care; ED, emergency department; OR, odds ratio; CI, confidence interval; RUB, resource utilization band. ′, ″ Splines. [a] For the 19—24 month follow-up period, diagnosis age was per 10 years. [b] Residential mobility included postal code changes in the 5 years prior to diagnosis. [c] Continuity of care included primary care clinician visits in the 6 to 30 months prior to diagnosis.

Diagnostic characteristics: Cancer site was strongly associated with UCC visits in all follow-up periods except the 13–18 month period. Individuals diagnosed with hematologic, lung, or breast cancers were significantly more likely to have a UCC visit compared to those diagnosed with other types of cancers during the first year of follow-up. The likelihood of a UCC visit increased over the follow-up periods for individuals diagnosed with genitourinary, lung, or digestive cancers but decreased for those diagnosed with breast cancer.

Sociodemographic characteristics: Age, sex, income quintile, and residential mobility were much less strongly associated with UCC visits. Individuals in the lowest income quintile were significantly more likely to visit the UCC in the first 6 months after diagnosis compared to those in the highest income quintile

Health care use history: Continuity of care, ED visits prior to diagnosis, PCC visits, and comorbidity were also less strongly associated with UCC visits.

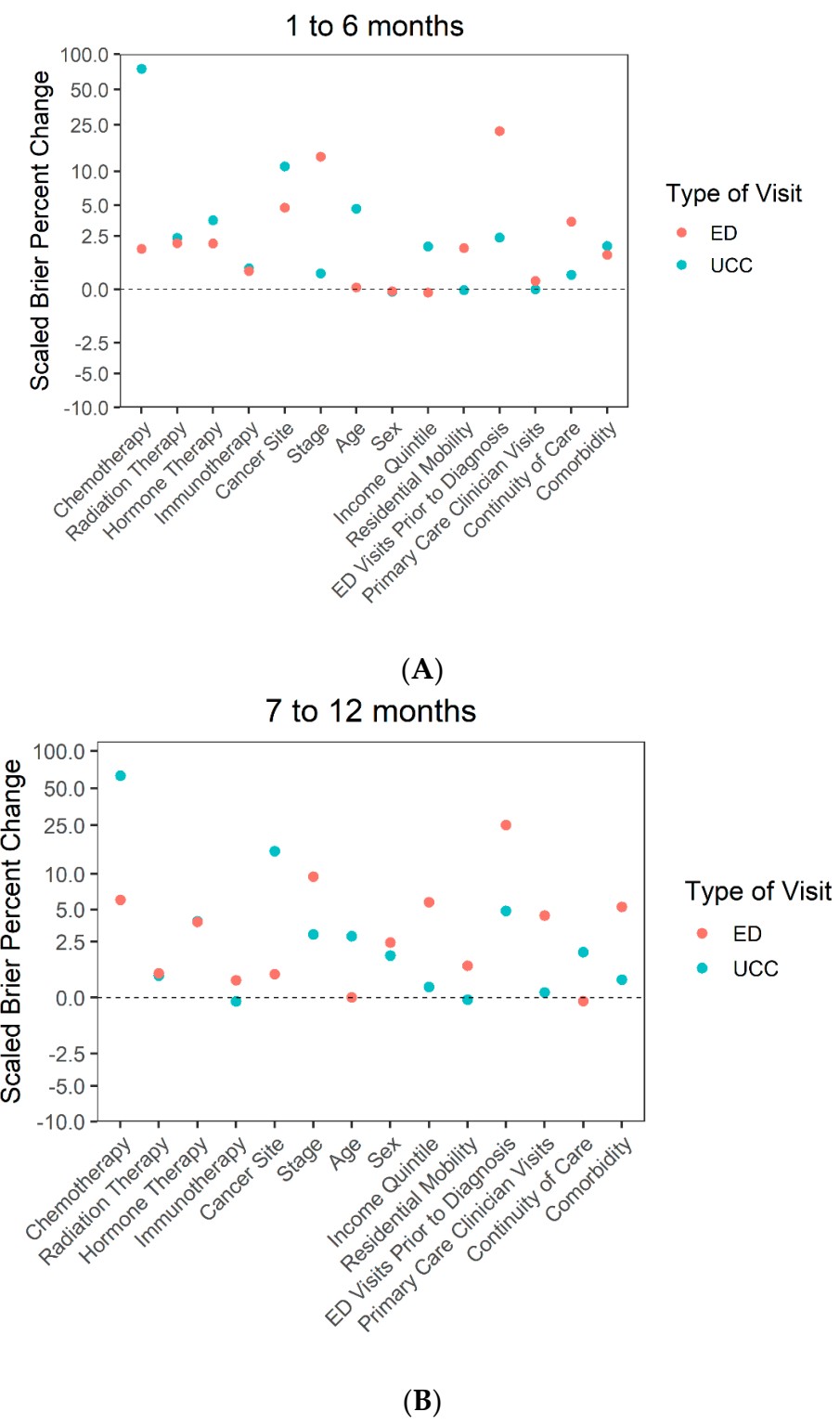

(**A**)

(**B**)

**Figure 2.** *Cont.*

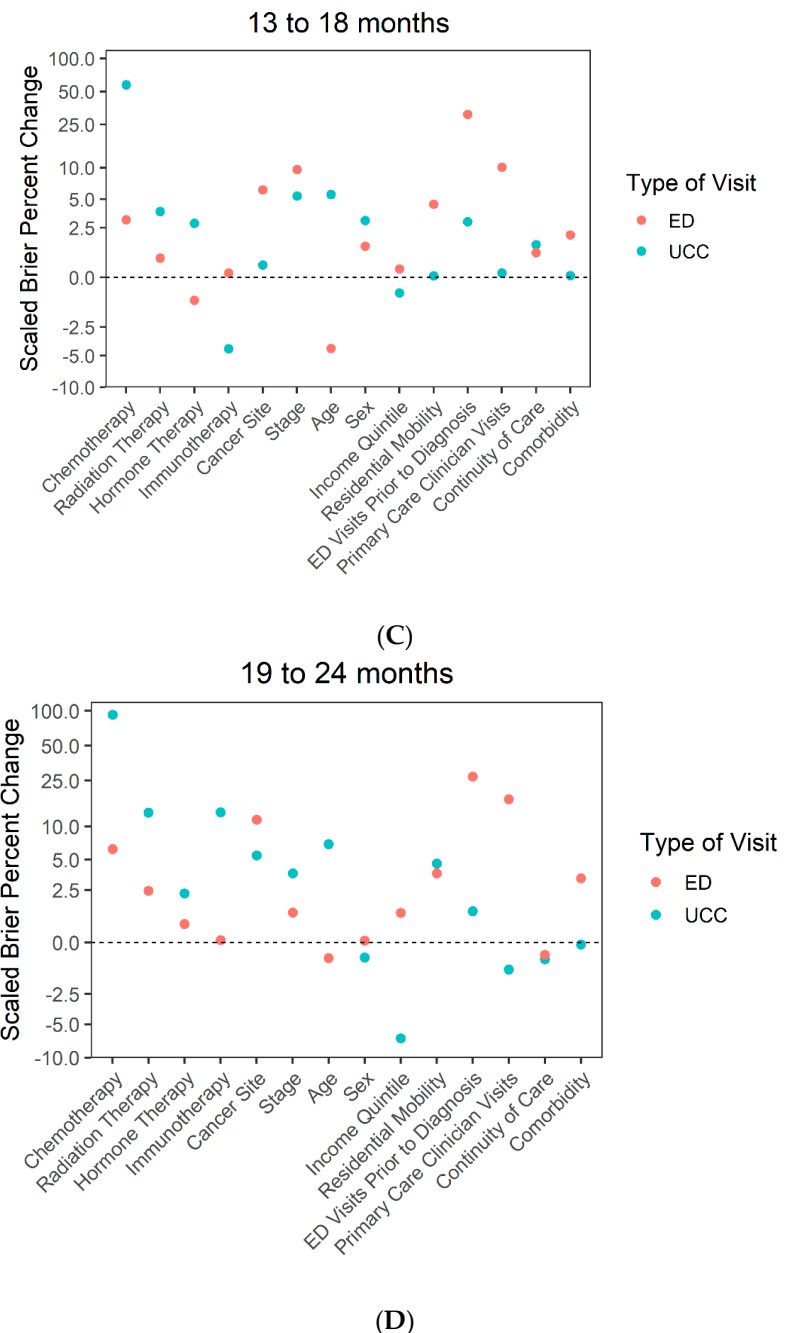

**Figure 2.** Impact of each predictor on UCC and ED visits by follow-up period, (**A**) 1 to 6 months, (**B**) 7 to 12 months, (**C**) 13 to 18 months and (**D**) 19 to 24 months.

*3.5. Association between Predictors and ED Visits*

Treatment variables: Chemotherapy, RT, HT, and immunotherapy were less strongly associated with ED visits than other factors.

Diagnostic characteristics: Stage at diagnosis was strongly associated with ED visits in the 1–6 month follow-up period, whereas cancer site was strongly associated with ED visits in the 19–24 month follow-up period. Individuals diagnosed at stage II, III, IV, or at an unknown stage were significantly more likely to visit an ED compared to those diagnosed at stage I during all follow-up periods (Table 3). Individuals that were diagnosed with lung or digestive cancers were also significantly more likely to have an ED visit compared to those diagnosed with other types of cancer.

**Table 3.** Multivariable logistic regression model describing factors associated with ED visits stratified by follow-up time period.

| Variable | Follow–Up Period | | | | | | | | | | | |
|---|---|---|---|---|---|---|---|---|---|---|---|---|
| | 1 to 6 Months | | | 7 to 12 Months | | | 13 to 18 Months | | | 19 to 24 Months | | |
| | OR | 95% CI | *p* | OR | 95% CI | *p* | OR | 95% CI | *p* | OR | 95% CI | *p* |
| Chemotherapy | | | | | | | | | | | | |
| Inactive | 1.00 | | | 1.00 | | | 1.00 | | | 1.00 | | |
| Active | 2.26 | 1.85–2.76 | <0.01 | 1.70 | 1.49–1.94 | <0.01 | 1.46 | 1.22–1.75 | <0.01 | 1.53 | 1.23–1.89 | <0.01 |
| None | 1.65 | 1.35–2.01 | | 1.22 | 1.05–1.41 | | 0.98 | 0.84–1.15 | | 0.93 | 0.78–1.10 | |
| Radiation Therapy | | | | | | | | | | | | |
| Inactive | 1.00 | | | 1.00 | | | 1.00 | | | 1.00 | | |
| Active | 0.95 | 0.77–1.18 | <0.01 | 0.71 | 0.56–0.90 | 0.02 | 1.16 | 0.87–1.53 | 0.50 | 1.34 | 0.93–1.93 | 0.21 |
| None | 1.43 | 1.22–1.67 | | 0.92 | 0.81–1.06 | | 0.98 | 0.84–1.14 | | 0.95 | 0.81–1.12 | |
| Hormone Therapy | | | | | | | | | | | | |
| Inactive | 1.00 | | | 1.00 | | | 1.00 | | | 1.00 | | |
| Active | 1.15 | 0.82–1.60 | <0.01 | 1.10 | 0.86–1.41 | <0.01 | 1.15 | 0.87–1.51 | <0.01 | 0.92 | 0.72–1.17 | <0.01 |
| None | 2.22 | 1.62–3.04 | | 2.32 | 1.92–2.81 | | 2.15 | 1.64–2.83 | | 1.46 | 1.14–1.87 | |
| Immunotherapy | | | | | | | | | | | | |
| Inactive | 1.00 | | | 1.00 | | | 1.00 | | | 1.00 | | |
| Active | 2.40 | 1.45–3.98 | <0.01 | 1.18 | 0.83–1.67 | 0.20 | 1.07 | 0.76–1.53 | 0.73 | 0.79 | 0.50–1.23 | 0.57 |
| None | 2.15 | 1.25–3.72 | | 1.38 | 0.96–1.97 | | 1.12 | 0.85–1.46 | | 0.95 | 0.75–1.21 | |
| Cancer site | | | | | | | | | | | | |
| Hematologic | 1.08 | 0.91–1.28 | | 1.02 | 0.79–1.33 | | 0.76 | 0.61–0.95 | | 0.91 | 0.68–1.20 | |
| Genitourinary | 1.04 | 0.86–1.27 | | 1.14 | 0.90–1.45 | | 1.09 | 0.87–1.35 | | 1.29 | 1.01–1.66 | |
| Lung and bronchus | 1.76 | 1.49–2.09 | <0.01 | 1.62 | 1.27–2.07 | <0.01 | 1.62 | 1.26–2.08 | <0.01 | 1.91 | 1.40–2.62 | <0.01 |
| Breast | 1.15 | 0.94–1.39 | | 1.38 | 1.04–1.84 | | 1.04 | 0.75–1.42 | | 0.80 | 0.57–1.13 | |
| Digestive | 1.65 | 1.38–1.96 | | 1.52 | 1.23–1.88 | | 1.50 | 1.17–1.91 | | 1.67 | 1.30–2.16 | |
| Other | 1.00 | | | 1.00 | | | 1.00 | | | 1.00 | | |
| Stage | | | | | | | | | | | | |
| I | 1.00 | | | 1.00 | | | 1.00 | | | 1.00 | | |
| II | 1.18 | 1.03–1.36 | | 1.32 | 1.10–1.58 | | 1.39 | 1.16–1.67 | | 1.31 | 1.06–1.62 | |
| III | 1.53 | 1.33–1.76 | <0.01 | 1.75 | 1.45–2.10 | <0.01 | 1.33 | 1.09–1.63 | <0.01 | 1.38 | 1.07–1.78 | <0.01 |
| IV | 2.36 | 2.04–2.73 | | 2.42 | 1.98–2.95 | | 2.15 | 1.76–2.62 | | 1.84 | 1.40–2.42 | |
| Unknown | 2.05 | 1.67–2.53 | | 1.70 | 1.35–2.12 | | 2.00 | 1.53–2.60 | | 1.69 | 1.26–2.27 | |
| Diagnosis age [a] | | | | | | | | | | | | |
| ′ | 1.05 | 1.01–1.08 | 0.01 | 0.93 | 0.65–1.32 | <0.01 | 1.08 | 0.75–1.54 | <0.01 | 1.34 | 0.94–1.91 | <0.01 |
| ″ | | | | 1.56 | 1.30–1.87 | | 1.77 | 1.41–2.22 | | 1.99 | 1.59–2.48 | |
| Sex | | | | | | | | | | | | |
| Female | 1.00 | | | 1.00 | | | 1.00 | | | 1.00 | | |
| Male | 1.04 | 0.94–1.14 | 0.45 | 1.30 | 1.15–1.46 | <0.01 | 1.18 | 1.04–1.35 | <0.01 | 0.97 | 0.82–1.16 | 0.76 |

**Table 3.** *Cont.*

| Variable | Follow–Up Period | | | | | | | | | | | |
|---|---|---|---|---|---|---|---|---|---|---|---|---|
| | 1 to 6 Months | | | 7 to 12 Months | | | 13 to 18 Months | | | 19 to 24 Months | | |
| | OR | 95% CI | *p* | OR | 95% CI | *p* | OR | 95% CI | *p* | OR | 95% CI | *p* |
| Income quintile | | | | | | | | | | | | |
| Q1 | 1.18 | 1.01–1.39 | | 1.47 | 1.21–1.77 | | 1.52 | 1.26–1.84 | | 1.56 | 1.26–1.94 | |
| Q2 | 1.09 | 0.94–1.27 | | 1.31 | 1.08–1.59 | | 1.32 | 1.08–1.63 | | 1.53 | 1.21–1.93 | |
| Q3 | 1.06 | 0.92–1.22 | 0.29 | 1.15 | 0.96–1.38 | <0.01 | 1.27 | 1.04–1.56 | <0.01 | 1.36 | 1.08–1.71 | <0.01 |
| Q4 | 1.02 | 0.91–1.16 | | 1.11 | 0.93–1.33 | | 1.39 | 1.15–1.69 | | 1.30 | 1.03–1.63 | |
| Q5 | 1.00 | | | 1.00 | | | 1.00 | | | 1.00 | | |
| Residential mobility [b] | | | | | | | | | | | | |
| Low | 1.00 | | | 1.00 | | | 1.00 | | | 1.00 | | |
| High | 1.45 | 1.24–1.70 | <0.01 | 1.28 | 1.00–1.64 | 0.05 | 1.56 | 1.23–1.98 | <0.01 | 1.51 | 1.12–2.05 | <0.01 |
| ED visits prior to diagnosis | | | | | | | | | | | | |
| 0 | 1.00 | | | 1.00 | | | 1.00 | | | 1.00 | | |
| 1 | 1.70 | 1.53–1.90 | <0.01 | 1.71 | 1.49–1.96 | <0.01 | 1.87 | 1.65–2.13 | <0.01 | 1.50 | 1.26–1.78 | <0.01 |
| 2+ | 2.39 | 2.12–2.70 | | 2.37 | 1.98–2.84 | | 2.46 | 2.02–3.00 | | 2.52 | 2.01–3.16 | |
| Primary care clinician visits | | | | | | | | | | | | |
| ′ | 1.16 | 0.99–1.36 | | 1.57 | 1.26–1.95 | | 1.10 | 0.76–1.59 | | 1.10 | 0.69–1.74 | |
| ″ | 0.98 | 0.62–1.54 | <0.01 | 2.16 | 1.18–3.94 | <0.01 | 1.79 | 1.36–2.37 | <0.01 | 2.35 | 1.73–3.21 | <0.01 |
| ‴ | 1.23 | 1.08–1.41 | | 1.65 | 1.36–2.00 | | 2.96 | 1.27–6.94 | | 3.18 | 1.20–8.48 | |
| ⁗ | | | | | | | 1.73 | 1.36–2.19 | | 2.38 | 1.83–3.10 | |
| Continuity of care [c] | | | | | | | | | | | | |
| Yes (≥50%) | 1.00 | | | 1.00 | | | 1.00 | | | 1.00 | | |
| No (<50%) | 1.04 | 0.94–1.15 | 0.11 | 1.09 | 0.93–1.27 | 0.31 | 1.24 | 1.05–1.46 | 0.04 | 1.04 | 0.87–1.25 | 0.74 |
| <3visits | 0.79 | 0.62–1.01 | | 0.85 | 0.64–1.15 | | 1.10 | 0.78–1.57 | | 1.14 | 0.77–1.69 | |
| Comorbidity (RUB) | | | | | | | | | | | | |
| 0–2 | 1.00 | | | 1.00 | | | 1.00 | | | 1.00 | | |
| 3 | 1.08 | 0.92–1.27 | <0.01 | 1.03 | 0.85–1.26 | <0.01 | 1.14 | 0.93–1.39 | 0.04 | 1.06 | 0.81–1.38 | <0.01 |
| 4–5 | 1.34 | 1.11–1.62 | | 1.40 | 1.10–1.79 | | 1.40 | 1.05–1.87 | | 1.45 | 1.07–1.95 | |
| Time interval | | | | | | | | | | | | |
| ′ | 0.31 | 0.25–0.37 | <0.01 | 0.99 | 0.97–1.02 | 0.67 | 1.03 | 1.00–1.06 | 0.08 | 1.02 | 0.98–1.06 | 0.30 |
| ″ | 0.90 | 0.82–1.00 | | | | | | | | | | |

Abbreviations: ED, emergency department; OR, odds ratio; CI, confidence interval; RUB, resource utilization band. ′, ″, ‴, ⁗ Splines. [a] For the 1–6 month follow-up period, diagnosis age was per 10 years. [b] Residential mobility included postal code changes in the 5 years prior to diagnosis. [c] Continuity of care included primary care clinician visits in the 6 to 30 months prior to diagnosis.

Sociodemographic characteristics: Age, sex, income quintile, and residential mobility were much less strongly associated with ED visits than other variables.

Heath care use history: ED visits prior to diagnosis was strongly associated with ED visits over all follow-up periods. Individuals that had one ED visit prior to diagnosis were significantly more likely to have an ED visit compared to those who had no prior visits. Those that had two or more ED visits prior to diagnosis were over twice as likely to have

an ED visit. PCC visits prior to diagnosis were strongly associated with ED visits in the 19–24 month follow-up period (Figure S2).

*3.6. Sensitivity Analyses*

Sensitivity analyses found no substantive difference from the primary results (Tables S4 and S5).

## 4. Discussion

*4.1. Main Findings*

Chemotherapy was the strongest predictor of UCC visits during all follow-up periods. Prior research has found that treatment is one of the main reasons why an individual diagnosed with cancer requires urgent or emergent care [6]. Cancer site was also a strong predictor of UCC visits; individuals diagnosed with lung cancer had higher odds of visiting the UCC regardless of time since diagnosis, whereas those diagnosed with breast or hematologic cancers had higher odds of visiting the UCC in the first year after diagnosis. In contrast, the strongest predictor of ED visits was a history of ED visits. This may reflect individuals accessing the health care system in the same manner prior to their cancer diagnosis or because they live close to an ED [28]. Stage at diagnosis was also an important predictor of ED use; the odds of an individual diagnosed with stage IV visiting the ED was double that for those diagnosed with stage I after adjusting for all other factors, including treatment.

We found that sociodemographic variables were much less strongly associated with UCC or ED but sometimes demonstrated differences in use (e.g., individuals in the lowest income quintile were significantly more likely to visit the UCC in the first 6 months after diagnosis compared to those in the highest income quintile). Because the UCC is located in a lower income area, individuals who live near the UCC may be more likely to visit. In the later follow-up periods, factors such as RT, cancer site, and PCC visits became important, but these results must be interpreted with caution because of limited power in the 19–24 month time period.

Our results also found that UCC visits increased over the first 4 months after diagnosis and then slowly decreased, whereas ED visits were highest immediately after diagnosis. This could be due to the need for CTAS level 1 (resuscitation) care that is not provided by the UCC early in the cancer journey. However, level 1 and level 2 (emergent) CTAS scores in the 1–6 month follow-up period were similar between the groups. It is more likely that individuals who visited the ED soon after diagnosis were not familiar with the services provided by the UCC, highlighting the need to ensure that individuals are aware of the UCC at or soon after diagnosis.

Lastly, we found large differences in the median time from triage to discharge between the UCC and EDs (2 versus 9 h). Moreover, the median time between UCC triage and arrival in the examination room was 1 h. These results support the idea that a UCC can provide timely care, which is particularly important for cancer patients.

*4.2. Comparison with Other Studies*

Urgent care clinics for individuals with cancer have been described in several studies from the United States and Korea [29,30], but these studies did not evaluate factors associated with UCC visits [31,32]. Studies have examined predictors of ED visits [12,33], factors associated with admission to hospital from the ED [8,34,35] or death [9], as well as descriptive studies of why individuals diagnosed with cancer visit an ED [6,7,36]. To our knowledge, this is the first study to examine predictors of both a UCC and an ED.

*4.3. Strengths and Limitations*

Although several studies and grey literature describe the implementation of UCCs, few evaluations have been published in peer-reviewed literature. Although the results are specific to Manitoba, they can be used by other jurisdictions to support the implementation

of UCCs. In addition to clinical data, we used data from previously validated population-based administrative health databases, and missing data were low [14,15,37,38]. In some cases, the UCC closed early because of capacity issues, and individuals were re-directed to an ED, which could have impacted the characteristics of individuals seen in the ED and the rate of visits. Although over 2 years of UCC clinic data were available, power was limited in the final follow-up period and therefore, results for this time period must be interpreted with caution. Because the analyses included multiple comparisons, some of the associations found could be due to chance and, as for all observational studies, the results may be prone to bias from unrecognized or unmeasured factors.

## 5. Conclusions

Factors strongly associated with UCC visits were related to treatment, whereas those strongly associated with ED visits were related to prior health care use. Sociodemographic factors were not strongly associated with UCC or ED visits. The UCC provides care more quickly than an ED. Our results can be used by other jurisdictions to support the implementation of UCCs. Future studies would be beneficial to plan service delivery and improve clinical outcomes and patient satisfaction.

**Supplementary Materials:** The following are available online at https://www.mdpi.com/article/10.3390/curroncol28030165/s1, Figure S1: Description of when individuals diagnosed with cancer were included in the study cohort, Figure S2: Percentage of individuals who had a UCC visit and an ED visit by follow-up month. Table S1: List of ED visits excluded from the analyses, Table S2: Hours between triage and discharge for individuals who visited the UCC or an ED by CTAS score, 2013–2016, Winnipeg, Manitoba (*N* = 13,252 visits), Table S3: Percent change in scaled Brier scores from full model for UCC and ED visits by variable and follow-up period, Table S4: Percent change in scaled Brier score from full model for UCC and ED visits by variable and follow-up period excluding sex from the model, Table S5: Percent change in scaled Brier score from full model for ED visits by variable and follow-up period only during UCC hours of operation and including all reasons for an ED visit.

**Author Contributions:** Conceptualization, K.D., M.P., B.G., H.S., M.K., T.F. and E.J.B.; data curation, P.L., K.G. and O.B.; formal analysis, P.L., K.G. and O.B.; funding acquisition, K.D., M.P. and H.S.; methodology, K.D., P.L., K.G., O.B., M.P. and H.S.; project administration, K.D.; supervision, K.D.; writing—original draft, K.D., P.L., K.G. and O.B.; writing—review and editing, K.D., P.L., K.G., O.B., M.P., B.G., H.S., M.K., T.F. and E.J.B. All authors have read and agreed to the published version of the manuscript.

**Funding:** This research was funded by the Canadian Institutes of Health Research (A02–151563).

**Institutional Review Board Statement:** The study was conducted according to the guidelines of the Declaration of Helsinki, and approved by the University of Manitoba's Health Research Ethics Board, Manitoba Health's Health Information and Privacy Committee, and CancerCare Manitoba's Research and Resource Impact Committee (project code: HS20816 (H2017:167), approval date: 24 05 17). Because data were de-identified, informed consent was not required.

**Informed Consent Statement:** Patient consent was waived because data were de-identified.

**Data Availability Statement:** The data that support the findings of this study are not publicly available to ensure and maintain the privacy and confidentiality of individuals' health information. Requests for data may be made to the appropriate data stewards (Manitoba Health, Seniors and Active Living's Health Information Privacy Committee, and CancerCare Manitoba's Research and Resource Impact Committee).

**Acknowledgments:** We gratefully thank the Canadian Institutes of Health Research for their support as well as CancerCare Manitoba, Manitoba Health, Seniors and Active Living, and the Winnipeg Regional Health Authority for the provision of data.

**Conflicts of Interest:** The authors declare no conflict of interest. The funders had no role in the design of the study; in the collection, analyses, or interpretation of data; in the writing of the manuscript; or in the decision to publish the results.

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
