# Peer review of "Predictors of Urgent Cancer Care Clinic and Emergency Department Visits for Individuals Diagnosed with Cancer"

_curroncol, doi:10.3390/curroncol28030165_

Round 1

Reviewer 1 Report

Thank you for the opportunity to review this article.

I began the review but I stopped because it was very difficult for me to understand the paper. Tables are abstruse. They show too many data and need clarification and synthesis. I do not understand the methodology. I understood that you compared 2 populations: cancer patients visiting an UCC and those visiting EDs but it is not clear for me how you selected them. Did you include all the cancer patients that visited UCC/EDs? How many EDs participated to the study 1 or 6? What “eligibility to visit” means. Is it the date of the first visit to UCC?

I do not understand either how you perform the logistic regression? You stated that you assessed the likelihood to visit an UCC or ED. But how did you proceed if all patients that you included visited the UCC or ED? And why did you stratify by follow-up time after the start of eligibility? This add difficulty to read the tables and understand the message.

Maybe you could more simply describe patients’ characteristics depending on the visit location, UCC or ED, without stratifying by follow-up time. I would be interested in comparing the chief complaint of patients visiting UCC and ED or their outcomes (discharged home, hospitalization). Plotting the time of UCC/ED visit depending on the time of diagnosis would be of interest and may enlighten when cancer patients are more likely to visit ED during their cancer follow up.

I also would be interested to know if cancer patients ED visits decreased when the UCC opened by comparing the curves of UCC and ED attendance by cancer patients over time

ABSTRACT

INTRO

L43: Add dyspnea to the « most common reasons for ED use by oncology patients”

L46: In the EPICANCER study, 1/3 of the oncology patients’ visits occurred between 6PM and 8AM. Besides, in your results, only 40% of the patients visited the ED during UCC hours of operation.

MATERIAL AND METHODS

L72: Is there a time limit to follow-up care from a CCMB clinic/provider. Please add it if this is the case. Do the clinic take care of oncologic patients whatever is the main symptom, even if there is no obvious link with cancer?

L74: what is the background of the FPO? Is he or she an oncologist or has an emergency curriculum?

How many patients arrives at the UCC/day? How many patients/year are attended in the EDs of your study? In your study, 3152 cancer patients visited the UCC in 3 years, that is to say less than 3 patients/day on average. Did the UCC received other patients that are not listed in your study? Please describe more precisely how this structure works. 3 patients /day may seem of little interest.

L79: Begin saying that it is a retrospective observational study and the objective. The study design and objectives are a little confusing. Please, clarify what is your goal and how did you proceeded

Please, shorten the Data sources part

L122: I don’t understand this sentence: “Continuity of care was measured by determining the individuals with at least 50% 122 of visits to the same PCC among those with at least three visits in six to 30 months prior to diagnosis (yes; >50%, no; <50%,<3 visits).”

L133: The information that “Non-cancer related ED visits were excluded” should arrive sooner”

L135: “Descriptive statistics were used to describe individuals who visited the UCC or ED.” This sentence is a pleonasm

RESULTS

Not sure that figure S2 is necessary

Table 1 is abstruse. To many data. Don’t stratify by follow-up time, give only one modality of a dichotomic variable (the other is implicit), don’t give median and mean, choose one.

Maybe you should merge figure 1A and 1B in order to compare the visit rate between UCC and EDs. Why the rate of ED visits decreased over time?

Figures with scaled brier percent are difficult to read. Please reverse x and y and order the variables from the highest to the lowest value (for UCC type of visit)

Tables 2 and 3 what means “inactive” chemotherapy, radiation, hormone…

Your data do not support the last sentence of your abstract and conclusion: “increased education about the UCC at diagnosis may increase…and potentially improve …cancer outcomes”

Reviewer 2 Report

Thank you for your work to conduct this interesting study and well-written manuscript. The study was well-developed, executed and analyzed, and the conclusions were relevant to describe utilization patterns in the geographical area of interest. For clarity, consider rephrasing the sentences in lines 66-68 as they seem misleading as currently written, implying that patients newly diagnosed with cancer may have high resuscitation needs, which even your data does not indicate as the case. 

Author Response

Thank you for taking the time to review this paper.  We appreciate your thoughtful comments and suggestions, which we believe have led to marked improvement in the revised manuscript. Please see our response in red.

Lines 66-68 have been revised as follows: The UCC is located at CCMB’s main site in the city of Winnipeg’s core area. During the study time period, there were six EDs located throughout Winnipeg including an ED located at the Health Sciences Centre, Manitoba’s largest academic health care facility.

Round 2

Reviewer 1 Report

thank you for your revised manuscript and the responses you provided